# Genomic comparison of diverse *Salmonella* serovars isolated from swine

**Sushim K. Gupta**[1], **Poonam Sharma**[1], **Elizabeth A. McMillan**[1,2], **Charlene R. Jackson**[1], **Lari M. Hiott**[1], **Tiffanie Woodley**[1], **Shaheen B. Humayoun**[1], **John B. Barrett**[1], **Jonathan G. Frye**[1☯‡]*, **Michael McClelland**[3☯‡]

**1** Bacterial Epidemiology and Antimicrobial Resistance Unit, USDA-ARS, Athens, GA, United States of America, **2** Department of Microbiology, University of Georgia, Athens, GA, United States of America, **3** Department of Microbiology and Molecular Genetics, University of California Irvine, Irvine, CA, United States of America

☯ These authors contributed equally to this work.
‡ These authors are joint senior authors on this work.
* Jonathan.Frye@ars.usda.gov

**Data Availability Statement:** Sequence data underlying this study are available from the NCBI Sequence Read Archive (SRA), with specific accession numbers listed in the Supporting

## Abstract

Food animals act as a reservoir for many foodborne pathogens. *Salmonella enterica* is one of the leading pathogens that cause food borne illness in a broad host range including animals and humans. They can also be associated with a single host species or a subset of hosts, due to genetic factors associated with colonization and infection. Adult swine are often asymptomatic carriers of a broad range of *Salmonella* servoars and can act as an important reservoir of infections for humans. In order to understand the genetic variations among different *Salmonella* serovars, Whole Genome Sequences (WGS) of fourteen *Salmonella* serovars from swine products were analyzed. More than 75% of the genes were part of the core genome in each isolate and the higher fraction of gene assign to different functional categories in dispensable genes indicated that these genes acquired for better adaptability and diversity. High concordance (97%) was detected between phenotypically confirmed antibiotic resistances and identified antibiotic resistance genes from WGS. The resistance determinants were mainly located on mobile genetic elements (MGE) on plasmids or integrated into the chromosome. Most of known and putative virulence genes were part of the core genome, but a small fraction were detected on MGE. Predicted integrated phage were highly diverse and many harbored virulence, metal resistance, or antibiotic resistance genes. CRISPR (Clustered regularly interspaced short palindromic repeats) patterns revealed the common ancestry or infection history among *Salmonella* serovars. Overall genomic analysis revealed a great deal of diversity among *Salmonella* serovars due to acquired genes that enable them to thrive and survive during infection.

## Introduction

*Salmonella enterica* subsp. *enterica* has the ability to infect a wide range of hosts, including both animals and humans. In the latter group, the bacterium causes foodborne illnesses

Information. All other relevant data are within the paper and its Supporting Information files.

**Funding:** JF and CJ were supported by United States Department of Agriculture (USDA) project plans 6040-32000-006-00 and 6040-32000-009-00. MM was supported in part by grants from the Foundation for Meat and Poultry Research and Education, and the National Cattlemen's Beef Association, and by the USDA grant 2017-67015-26085 on which MM is a co-PI.

**Competing interests:** The authors have declared that no competing interests exist.

ranging from mild diarrhea and gastroenteritis to severe systemic infections such as enteric fever [1], second only to norovirus as the causative agent of foodborne illness; *Salmonella* infections in humans are the leading cause of hospitalization and deaths from foodborne illness in the United States [2]. The Centers for Disease Control and Prevention has estimated 1.2 million foodborne illnesses with 19,000 hospitalizations and approximately 380 deaths in the United States annually (https://www.cdc.gov/salmonella/index.html). *S. enterica* ranks as the leading cause of foodborne disease as measured by the combined cost of illness and quality-adjusted life-year [3]. The direct economic losses owing to salmonellosis in the US exceed an estimated $3.5 billion per year. More than 2,600 *Salmonella* serovars has been confirmed by the agglutination properties of the somatic O, flagellar H, and capsular Vi antigens [4].

Inappropriate use of antimicrobials in food animal production during treatment and prevention of diseases and for growth promotion contribute to resistance, including acquisition of antibiotic resistance (AR) genes through horizontal gene transfer (HGT). Multi drug resistant (MDR) bacteria are recognized as a major threat to public health [5] and require a comprehensive approach to combat them [6]. Resistance determinants are often present on mobile genetic elements (MGE), such as plasmids, integrons etc. and can be transferred among multiple bacterial genera [7, 8]. A wide range of plasmids that carry AR and virulence genes have been reported in *Salmonella* [9–11]. Here we catalog all the antibiotic resistance genes and their organization in a diverse set of *Salmonella* from swine.

Phage play a profound role in bacterial evolution as they assist in transfer of antibiotic resistance (AR) and, virulence genes, including inserting their genome into the host's DNA [12]. These lysogens can also become lytic by replicating and then killing their temporary host bacteria. A large number of phage have been reported from *Salmonella*, such as Fels-1, Gifsy-2, P22, FelixO1, etc., and some of them carry several virulence and resistance gene cassettes [5, 13–16]. Here, we investigate phage distribution among *Salmonella* serovars to determine the resistance and virulence genes associated with phage. Here we also characterize the CRISPR (clustered regularly interspaced short palindromic repeats) elements, which act as an adaptive immune system against exogenous DNA [17], including phage DNA. In addition, analysis of CRISPR sequences improve the discriminatory power of molecular characterization of *Salmonella* [18].

Few studies have been conducted to identify genetic factors associated with *Salmonella* serovars isolated from swine using WGS [19, 20]. In order to understand the genetic variations among different *Salmonella* serovars, the WGS of fourteen *Salmonella* serovars isolated from swine and swine swab were analyzed. The WGS data was assessed for resistance and virulence determinants, and their association with MGE was predicted. WGS was further analyzed to identify CRISPR elements and phage associated resistance and virulence genes.

## Material and methods

### Isolate selection and antimicrobial susceptibility testing

Fourteen *Salmonella* isolates were collected by the National Antimicrobial Resistance Monitoring System (NARMS) in year 2004–2005. All isolates were streaked onto Mueller-Hinton (MH) agar (Oxoid, Cambridge, UK). A single colony was selected and subsequently inoculated in MH broth (Oxoid, Cambridge, UK), and incubated for 16–18 h at 37°C with shaking (250 rpm). All isolates were subjected to susceptibility testing via the Sensititre™ semi-automated antimicrobial susceptibility system (TREK Diagnostic Systems, Inc.) using a custom-made panel including amikacin, gentamicin, kanamycin, streptomycin, ampicillin, amoxicillin-clavulanic acid, ceftiofur, ceftriaxone, cefoxitin, sulfamethoxazole/sulfisoxazole, trimethoprim-sulfamethoxazole, chloramphenicol, ciprofloxacin, nalidixic acid, and tetracycline [21]. The

isolates were subjected to preliminary biochemical screening to distinguish the different serogroups using serogroup-specific antisera (Difco Laboratories, Detroit, MI) and serotyping was used to identify serovars at the National Veterinary Services Laboratories, APHIS, USDA (Ames, IA).

## Genome sequencing and analysis

Genomic DNA was isolated using GenElute bacterial genomic DNA kit (Sigma-Aldrich, St. Louis, MO) according to manufacturer's protocols. The quality of DNA was tested using Nanodrop and quantified using an Invitrogen Qubit 2.0 Fluorometer (Life Technologies, Carlsbad, CA). The extracted DNA was stored in 1X TE buffer (pH 8.0) at -20˚C until further use. Paired-end sequencing was performed on an Illumina HiSeq2500 (Illumina Inc., San Diego, CA) as described elsewhere [22] and the reads were deposited in the SRA database under the bioproject PRJNA254816. The SRA accession numbers is available in Table 1. The reads were retrieved from the SRA dataset and reads with a phred score ≥30 were *de novo* assembled using the A5-pipeline_version_20141120 [23]. The genome and plasmid contigs from the assembly were sorted on the basis of remote blast using an in-house python script. The genomic contigs were annotated using PROKKA [24]. In order to reconstruct the pangenome the assembled genomes were annotated with Prokka and used as input for Roary with the identity cut-off of 95% [25]. Roary generated clusters of homologous gene groups from which core, accessory and unique genes were predicted. The Clusters of Orthologous Groups of proteins (COGs) database was used for the functional annotation [26]. The amino acid sequences generated from the Prokka was used as input for functional annotation based on orthologous group using WebMGA online server (http://weizhong-lab.ucsd.edu/meta genomic-analysis/server/cog/). The assembled contigs and deduced amino acid sequences from Prokka were used to predict the acquired antibiotic and metal resistance genes using the ARG-ANNOT [27] and BacMet [28] respectively, with e-value (1e -10), coverage ≥90% and identity ≥90%. The adherence-associated gene cluster were identified using the Virulence Factor Database (VFDB) [29] using a threshold of ≥95% identity and ≥95% coverage. Lysogenic phage were predicted using the web-based tool PHASTER [30]. CRISPR regions were predicted using CRISPR finder [31] and blastn was done to identify similar phage among

**Table 1. Assembly statistics of the *Salmonella enterica* serovars isolated from swine and their products.**

| *Salmonella* serovar | Strain | Year | Genome | | | | | No of CDS | SRA_ID |
|---|---|---|---|---|---|---|---|---|---|
| | | | Length (b) | No of Contigs | Coverage(x) | GC (%) | N50 | | |
| Agona | CRJJGF_0019 | 2005 | 4933904 | 101 | 78 | 52.1 | 162691 | 4,680 | SRX791372 |
| Anatum | CRJJGF_0121 | 2004 | 4707212 | 65 | 76 | 52.1 | 156274 | 4,414 | SRX791374 |
| Bovismorbificans | CRJJGF_0070 | 2004 | 4662088 | 89 | 93 | 52.1 | 168471 | 4,354 | SRX791423 |
| Choleraesuis | CRJJGF_0148 | 2004 | 4708469 | 99 | 80 | 52.1 | 149708 | 4,528 | SRX791500 |
| Cubana | CRJJGF_0088 | 2004 | 4929533 | 78 | 58 | 52.1 | 155115 | 4,604 | SRX791441 |
| Give | CRJJGF_0073 | 2004 | 4613042 | 61 | 89 | 52.1 | 226100 | 4,308 | SRX791426 |
| Heidelberg | CRJJGF_0002 | 2004 | 4853670 | 123 | 92 | 52.1 | 158927 | 4,558 | SRX791355 |
| Infantis | CRJJGF_0031 | 2005 | 4650782 | 75 | 76 | 52.1 | 157448 | 4,351 | SRX791384 |
| Minnesota | CRJJGF_0078 | 2004 | 4651740 | 87 | 95 | 52.1 | 372885 | 4,352 | SRX791431 |
| Manhattan | CRJJGF_0112 | 2005 | 4668352 | 201 | 90 | 52.1 | 260620 | 4,361 | SRX791465 |
| Ohio | CRJJGF_0161 | 2005 | 4854746 | 91 | 74 | 52.1 | 216733 | 4,532 | SRX791512 |
| Tennessee | CRJJGF_0089 | 2004 | 4772495 | 92 | 53 | 52.1 | 164659 | 4,462 | SRX791442 |
| Typhimurium | CRJJGF_0051 | 2004 | 4939221 | 130 | 69 | 52.1 | 172339 | 4,667 | SRX791404 |
| Worthington | CRJJGF_0141 | 2004 | 4848597 | 97 | 75 | 52.1 | 210218 | 4,559 | SRX791493 |

*Salmonella* serovars and linear comparison of similar phage was done using Easyfig [32]. The Integrall integron database (http://integrall.bio.ua.pt) was used to analyze and assign integron sequences [33]. Genes associated with plasmid replicon were identified using PlasmidFinder to identify the target sequence in the genomes of each isolate [34].

## Results

This study was the part of retrospective study to maximize the understanding of the AR gene distribution, MGE and genome diversity of the *Salmonella* [22] from food animals. Fourteen *Salmonella* isolates from swine were selected based on differences in phenotypic AR profile, serotype from the collection of NARMS isolates between 2004 and 2005.

### General features and sequenced genomes

The genome statistics are presented in Table 1. In brief, the average number of contigs per genome was 95 (range: 61 to 201 contigs). The median assembly coverage ranged between 53 to 95 fold which was adequate to produce bacterial draft genomes [35]. The average GC content of each genome was 52.1% which was consistent with that of the complete *S. enterica* chromosome [36]. The average number of coding sequences (CDSs) per isolate was 4,480 and the highest and lowest number of CDS was obtained for *S.* Agona (4,680) and *S.* Give (4,308), respectively. Additionally, average nucleotide identity matrix showed >98% identity among the serovars and >99.9% identity to their respective reference genome sequences from NCBI Gen-Bank. The high average nucleotide identity (ANI) [37] values of pair wise genome comparisons again confirmed that these serovars were nearly identical to their corresponding reference serovars. The reads were deposited in the GenBank Sequence Read Archive (SRA) database and details are provided in Table 1.

### Antimicrobial resistance genes, integrons, and plasmids

Antibiotic susceptibility assays were performed according to Clinical and Laboratory Standards Institute (CLSI) standards. Eleven isolates were resistant to at least one of the tested antibiotics and nine of them were multidrug resistant (MDR; resistant to two or more antimicrobials); the remaining three *Salmonella* serovars Give, Infantis, and Manhattan were susceptible to all tested antimicrobials. None of the isolates were resistant to azithromycin, ciprofloxacin or nalidixic acid. The susceptibility testing results and corresponding predicted AR genes results are summarized in Table 2. High frequency of resistance to tetracycline (9/14, 64.3%) was observed among the isolates, followed by resistance to beta-lactams (7/14, 50.0%), aminoglycosides (7/14, 50.0%) and sulfonamides (7/14, 50.0%). Three isolates 3/14, 21%). (*S.* Agona, *S.* Ohio and *S.* Typhimurium) were resistant to two antibiotics (chloramphenicol and sulfamethoxazole) while two isolates (2/14, 14.3%); *S.* Anatum and *S.* Worthington) were only resistant to tetracycline. The correlation between phenotypically confirmed AR and *in silico* predicted AR genes of the serovars is presented in Table 2. Resistance determinants were predicted with ARG-ANNOT. Hits with > 90% identity and 100% coverage were considered as positive resistance determinants. A total of 73 AR genes (59 in Table 1 and 14 *aac6-I* variants) were predicted from the genome sequences and at least one AR gene was predicted in each isolate (Table 1). The aminoglycoside resistance gene variants Y and aa of aac6-I was detected in all the isolates and these cryptic variants were the only aminoglycoside resistance gene variants predicted in *Salmonella* serovars Give, Infantis, and Mannhattan exhibited no phenotypic resistance. Among eight streptomycin resistant isolates *strA* and *strB* were detected in five isolates while four isolates carry *aadA* gene variants and all these three genes (*strA*, *strB*, and *aadA*) were detected in *S.* Heidelberg. Seven isolates were resistant to beta-lactams and three

**Table 2. Antibiotic resistance phenotype and predicted antibiotic resistance and virulence genes in different *Salmonella* serovars.**

| *Salmonella* serovar | Virulence Gene | | AGly | Bla | Tet | Sul | Chl | Others | Replicons |
|---|---|---|---|---|---|---|---|---|---|
| **Agona** | *lpfA/B/C/D/E; avrA, slrP, sspH2, sseK1* | G* | *aph3"-Ia, strA, strB* | *bla*<sub>CMY-2</sub> | *tet (A)* | *sul1, sul2* | *flo (R)* | *dfrA7/fosA* | *incA/C2* |
| | | P** | STR | AMP, AUG, AXO, FOX, TIO, COT | TET | FIS | CHL | – | |
| **Anatum** | *avrA, slrP, sspH2* | G | – | – | *tet (C)* | – | – | – | *colE10* |
| | | P | – | – | TET | – | | | |
| **Bovismorbificans** | *lpfA/B/C/D/E, avrA, ratB, slrP, sspH2, sodCI, sseK1* | G | *aph4-Ia, aac-Iva, strA, strB* | *bla*<sub>TEM-1</sub> | *tet (B)* | *sul1* | – | – | *IncHI2 incHI2A* |
| | | P | GEN, STR | AMP, | TET | FIS | | | |
| **Choleraesuis** | *lpfA/B/C/D/E; gogB, ratB, slrP, sspH2, sodCI, sseK1* | G | *strA, strB* | – | *tet (B)†* | *sul1* | – | – | *Col440II, incFIB, incQ1* |
| | | P | STR | – | – | FIS | | | |
| **Cubana** | *lpfA/B/C/D/E; avrA, slrP, sspH2, sseK1* | G | *aadA7* | – | *tet (A)* | *sul1* | – | – | – |
| | | P | STR | – | TET | FIS | | | |
| **Give** | *ratB, slrP, cdtB* | G | – | – | – | – | – | – | – |
| | | P | – | – | – | – | | | |
| **Heidelberg** | *lpfA/B/C/D/E; avrA, ratB, slrP, sspH2, sodCI* | G | *stra, strB, Aph4-Ia, aac-Iva, aadA2, sph, aphA2* | *bla*<sub>TEM-1</sub> | *tet (B)* | *sul3* | – | *dfrA12/ fosA* | *incFII, incHI2, incHI2A* |
| | | P | GEN, STR | AMP, COT | TET | FIS | – | | |
| **Infantis** | *lpfA/B/C/D/E; avrA, ratB, slrP, sspH2, sseK1* | G | – | – | – | – | – | – | *incI1* |
| | | P | – | – | – | – | | | |
| **Minnesota** | *avrA, ratB, cdtB, slrP, sseK1* | G | – | *bla*<sub>CMY-2</sub> | *tet (A)* | – | – | – | *incI1, incP6* |
| | | P | – | AMP, AUG, AXO, FOX, TIO | TET | – | | | |
| **Manhattan** | *lpfA/B/C/D/E; avrA, ratB, slrP, sspH2* | G | – | – | – | – | – | – | – |
| | | P | – | – | – | – | | | |
| **Ohio** | *ratB, sspH2, sseK1* | G | *strA, strB, aph3"Ia* | *bla*<sub>TEM-1</sub>, *bla*<sub>CMY-2</sub> | *tet (A)* | *sul1, sul2* | *flo (R)* | *dfrA1* | *incA/C2* |
| | | P | STR | AMP, AUG,AXO, FOX, COT, TIO | TET | FIS | CHL | | |
| **Tennessee** | *lpfA/B/C/D/E; avrA, slrP, sspH2, sseK1* | G | *aadA2* | *bla*<sub>CARB-2</sub> | – | – | – | *DfrA16/ere (A)/fosA* | *incN* |
| | | P | STR | AMP, COT | – | – | – | | |
| **Typhimurium** | *lpfA/B/C/D/E; pefA/B/C/D; spvB/C/R; avrA; gogB, ratB, slrP, sspH2, sodCI, sseK1* | G | *aadA2,* | *bla*<sub>CARB-2</sub> | *tet (G)* | *sul1* | *floR* | – | *incFIB* |
| | | P | STR | AMP | TET | FIS | CHL | | |
| **Worthington** | *lpfA/B/C/D/E; avrA, slrP, sspH2, sseK1* | G | – | – | *tet (B)* | – | – | – | *incI1* |
| | | | | | | | | | |
| | | P | – | – | TET | – | | | |
| No of ARG | | | 22 | 8 | 9 | 9 | 3 | 8 | |

*: Predicted gene

**: Confirmed phenotype

†: Partial/truncated gene

**Antibiotic used**- AMP: Ampicillin, AUG: Augmentin, AXO: Ceftriaxone, AMX: Amoxicillin, AZM: Azithromycin, COT: Cotrimoxazole, CHL: Chloramphenicol, ERY: Erythromycin, FIS: Sulfisoxazole, FOX: Cefoxitin, GEN: Gentamicin, STR: Streptomycin, TET: Tetracycline, TIO: Ceftiofur

**Antibiotic classes**- AGly: Aminoglycosides, Bla: Betalactamases, Tet: Tetracycline, Sul: Sulfonamides, Chl: Chloramphenicol

genes encoding beta-lactamases were identified in these seven isolates, with the most common being $bla_{TEM-1}$ ((3/14), 21.4%) and $bla_{CMY-2}$ ((3/14), 21.4%), followed by $bla_{CARB-2}$ ((2/14), 14.3%). The Amber class A β-lactamase gene ($bla_{CARB-2}$) conferring resistance to ampicillin were detected in serovar Tennessee and Typhimurium. Amber classA potential ESBL (extended spectrum of beta-lactamase) gene ($bla_{TEM-1}$) conferring resistance to ampicillin was detected in serovars Bovismorbificans, Heidelberg and Ohio. Amber class C genes ($bla_{CMY-2}$) that conferred resistance to ampicillin, amoxicillin, and ceftiofur were detected in serovars Agona, Minnesota and Ohio. Four different tetracycline resistance gene allele's *tet*A, *tet*B, *tet*C and *tet*G were identified in the analyzed *Salmonella* serovars. With *tet*A and *tet*B the most frequently detected tetracycline resistances, occurring in 4/14, 28.6% and 3/14, 21.4% of isolates, respectively, followed by *tet*C and *tet*G each in 1/14, 7.1% of isolates. The macrolide resistance gene *ere*A was detected in Tennessee, while trimethoprim gene variants (*dfr*A1, A7, A12, and A16) were detected in serovars Ohio, Agona, Heidelberg, and Tennessee respectively. The sulfonamide resistance gene alleles were identified in seven *Salmonella* serovars; only *sul1* was detected in serovar Bovismorbificans, Choleraesuis, Cubana and Typhimurium ((4/14); 28.6%), while *sul1* and *sul2* were detected in serovar Agona and Ohio ((2/14); 14.3%) and *sul3* was detected in serovar Heidelberg ((1/14); 7.1%). The *floR* gene was detected in three of fourteen chloramphenicol resistant isolates and only these three serovars Agona, Ohio, and Typhimurium conferred resistance to chloramphenicol. No mutations in chromosomal genes *gyrA*, *gyrB*, *parE*: conferring resistance to ciprofloxacin and nalidixic acid and *rRNA* genes 23S *rRNA*, *rplD*, *rplVb* conferring resistance to macrolide were observed; a point mutation in *parC* at position T57S was observed in most of the studied serovars except Bovismorbificans and Typhimurium; however, none of these isolates were resistant to nalidixic acid (S2 Table).

Class 1 integrons were detected in six isolates that were grouped into five different integron profiles (*ln127*, *in167*, *in363*, *in1581*, and *ln1582*). Most of these integrons carried a quaternary ammonium compound resistance gene, *qacE*, and a sulfonamide resistance gene, *sul1*, as additional genes after the 3'end of the conserved segment. The class 1 integron genesin different *Salmonella* serovars are summarized in Table 3.

The presence of plasmids in the assembled contigs were confirmed by *in silico* replicon typing. The replicon type *IncA/C2* was identified in MDR serovars Agona and Ohio and both these isolates harbored identical class 1 integrons *In363*. The replicon type *IncHI2* was detected in serovars Bovisformicans and Heidelberg while *IncI1* was identified in serovars Infantis, Minnesota and Worthington. The replicon type *IncFII* was identified in serovars Choleraesuis, Heidelberg and Typhimurium while *IncN* and *IncP6* were detected in serovars Tennessee, and Minnesota, respectively. No replicon types were identified in serovars Cubana, Give, or Manhattan. Additional replicon *col440II*, and *IncQ* was identified in *S*. Choleraesuis. Eight different

**Table 3. Class 1 integrons identified in different *Salmonella* isolates.**

| *Salmonella* serovar | Strain name | Integron genes | Integron number | Other resistance and virulence genes on integron |
|---|---|---|---|---|
| Agona | CRJJGF_0019 | *IntI1-dfrA1-attC—gcuC-attC-3'CS* | *In363* | *sul1, qacE,* |
| Cubana | CRJJGF_0088 | *IntI1-aadA7-attC-3'CS* | *In1581* | *sul1, qacE* |
| Heidelberg | CRJJGF_0002 | *IntI1-dfrA12-attC—gcuF—attC—aadA2-attC-3'CS* | *In127* | *qacE** |
| Ohio | CRJJGF_0161 | *IntI1-dfrA1-attC-gcuC-attC-3'CS* | *In363* | *sul1, qacE* |
| Tennessee | CRJJGF_0089 | *IntI1-dfrA16c-attC- bla$_{CARB-2}$-attC-aadA2-attC-ereA1c-attC3'CS* | *In1582* | *qacE** |
| Typhimurium | CRJJGF_0051 | *IntI1-bla$_{CARB-2}$—attC-3'CS* | *In167* | *qacE** |

*: Partial gene

resistance gene clusters were detected in different *Salmonella* serovars and are presented in Fig 1.

The AR genes clusters were highly diverse, with the exception of a $> 35$ kb (99.9% identical) region of homologous genes carrying *tet*A, *str*A, *str*B, *sul*2, and *bla*$_{CMY-2}$, which were detected in *Salmonella* serovar Agona and Ohio. The region of homologous genes carrying *strA/strB*, and *strA/strB/sul2* were detected in *Salmonella* serovar Bovismorbificans and Choleraesuis respectively, while *aad*A/*ebr*/*sul1* and *ereA/aadA2/bla*$_{CARB-2}$/*dfrA16* were detected in *Salmonella* serovar Cubana and Tennessee respectively. The gene clusters *aphA2/sph*, and *sul1/floR/tetA* were detected in *Salmonella* serovar Heidelberg and Typhimurium respectively.

## The pan-genome and functional genes comparison

Pangenome analysis was initiated using 62,730 genes of 14 *Salmonella* serovars that resulted into 8,174 clusters, of these 3,456 (42.3%) and 2,201 (26.9%) clusters were part of the core and accessory gene respectively. The remaining 2,517 (30.8%) independent genes identified as the unique genes. The core, accessory and unique genes are represented as inner circle, outer circle, and petals in the floral diagram respectively (Fig 2) and the details of core, accessory and unique genes in each serovars is given in S1 Fig.

The unique genes among serovars ranged from 93 to 268, with highest and lowest genes in serovars Worthington and Bovismorbificans respectively. The functional distribution of genes among serovars were examined using the COG database [26]. Functions encoded by the genes in these serotypes revealed $> 75\%$ of predicted ORFs were assigned to 24 COG functional group (Fig 3).

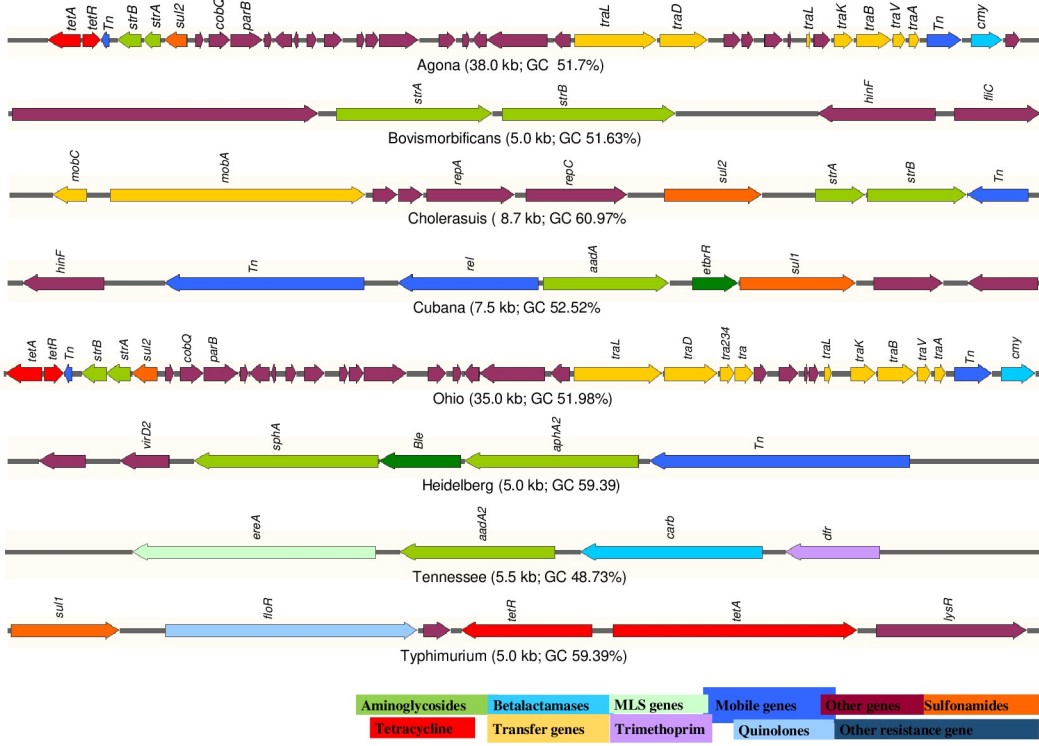

**Fig 1. Antibiotic resistance gene clusters in different *Salmonella* serotypes.** Bleomycin and EtBr resistance gene (Green) flanked by antibiotic resistance genes marked in green. The arrow shows orientation of the genes and genes are color coded to define different classes of antibiotic resistance, mobile and other genes categories.

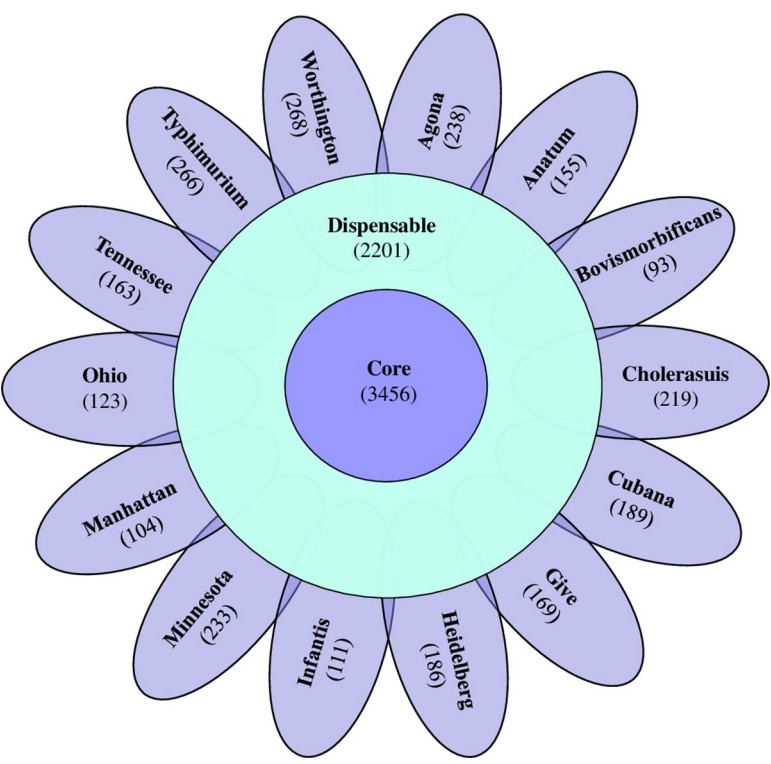

**Fig 2. Floral venn diagram showing the pangenome of 14 *Salmonella* serovars from swine.** The orthologus genes identified in all serovars presented in the center as core genes, orthologus genes identified among the serovars but not in all serovars presented in the periphery as accessary genes and each petals represents the unique genes in respective serotypes.

The most abundant COG functional group among serovars was the transport and metabolism of carbohydrates (G) and the highest (426) and lowest number of genes (385) in this category was noticed in *S.* Typhimurium and *S.* Minnesota respectively. The next abundant COGs were Transcription (K) followed by amino acid metabolism and transport (E). The highest number of genes in K and E category were noticed in *S.* Worthington (346) and *S.* Ohio (381) respectively, however the lowest number of genes in K (322) and E (370) COGs were noticed in *S.* Minnesota. A single COG functional gene was noticed in all the *S.* serotypes from extracellular structure categories. The distribution of functionally characterized COG genes in core and accessory genome in *Salmonella* serovars (Fig 4) revealed that the functional genes in the core genome ranged 76% to 82%.

Uniform distributions of functional genes from different COG category was observed for core and accessory genes among *Salmonella* serovars. However, the percent abundances of genes in category G (carbohydrate transport and metabolism; 26–28%) and L (Replication, recombination and repair 21–23%) among accessory genomes were greater among *Salmonella* serotypes compared to other functional categories. The core genomes in each *S.* serotype were commonly enriched in COG categories D, F, H, I, J, O and Q relative to those seen in the accessory genomes.

## Virulence genes, phage and CRISPR in *Salmonella* serovars

Genes with >95% sequence homology and coverage when compared to the virulence factor database (VFDB) were considered matches in this analysis. The complete virulence gene profiles of each *Salmonella* isolate is shown in S1 Table and some of the gene variants that were

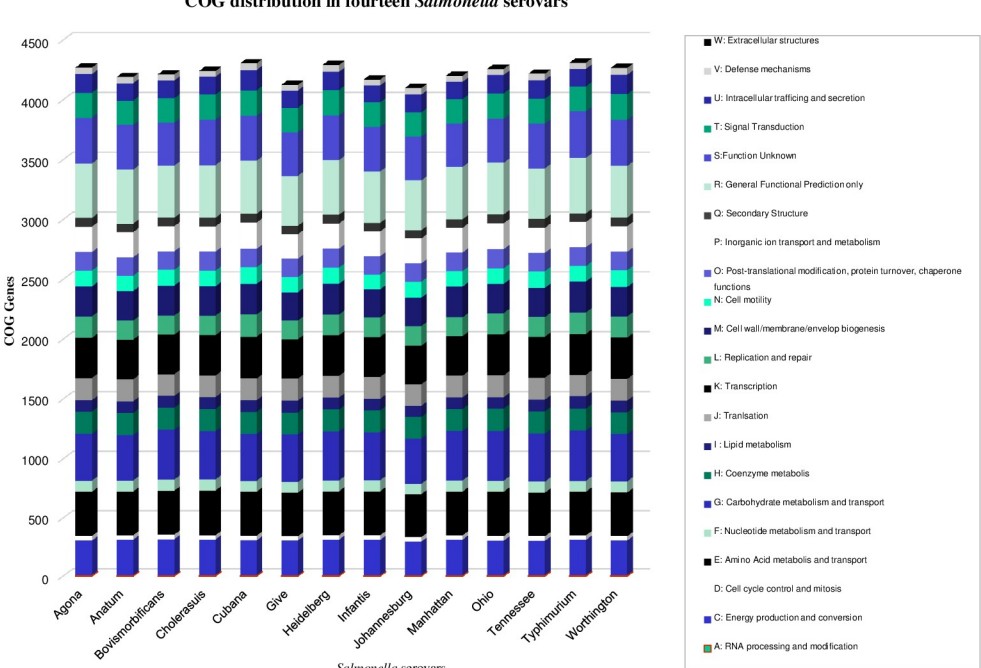

**Fig 3. Distribution of functional classes of predicted genes according to the clusters of orthologous groups in *Salmonella* serovars.** Different colors define different COG categories of genes.

not detected in all the isolates is shown in Table 2. The majority of the virulence genes were detected in all the *Salmonella* serovars. These virulence genes were very similar, with small variations in nucleotide/amino acid sequences, while few virulence genes were missing in single or multiple *Salmonella* serovars. The fimbrial adherence gene operons *bcfA/B/C/D/E/F/G*, *fimI/C/D/H*, *invA/B/C/E/F/G/H/I/J*, *csgBA* and *csgD/E/F/A* were the most common. Most of the isolates were positive for the long polar fimbrial operon *(lpfA/B/C/D/E)* except *Salmonella* serovars Anatum, Give, Minnesota, and Ohio; however, *lpfD* was highly diverse among all of the *Salmonella* isolates. The plasmid-encoded fimbrial operon (*pefA/B/C/D*) and *Salmonella* plasmid virulence (*spvB/C/R*) genes were detected in *S*. Typhimurium and none of these genes were detected in other isolates. The pathogenicity island 1 (SPI-1), encoding type III secretion system (T3SS) secreted effector genes *sipA/B/C/D*, *sopA*, *sopB*, *sopE2*, and *misL*, were highly similar in all the isolates with the exception of s*ipD* genes that shared 91% similarity to the reference gene (*S*. Typhimurium; NP_461804) in *S*. Minnesota. *AvrA* was identified in all the isolates except *S*. Choleraesuis and *S*. Ohio. The SPI-2 encoded T3SS genes, including *spiC*, *sifA*, *sifB*, *sseF*, *sseG*, *sseL*, *pipB*, *pipB2*, *sopD2*, and *slrP*, were detected in all the isolates. *GogB* was only detected in *Salmonella* serovars Choleraesuis and Typhimurium. Other effector genes, including *ratB*, *sseK1*, *sseK2*, and *sspH2*, were detected in some of the *Salmonella* serovars (S1 Table). The *sspH2* effector homolog was divergent (85.52% identical) in *S*. Ohio. The sensory systems genes *phoP/Q* were identical in all the serovars with an exception in *S*. Worthington where a secondary mutation was observed at nucleotide position 450 (A-T) with coverage depth >50x.

Genome contents were further compared, and it was observed that several genes were confined to one serovar, or were highly diverse even if they were annotated as the same gene (S3 Table). For example a 27 kb conting in *S*. Agona harboring subset of type VI secretion system (T6SS). A unique pathogenesis protein, *kcpA*, was identified on a phage in *S*. Anatum.

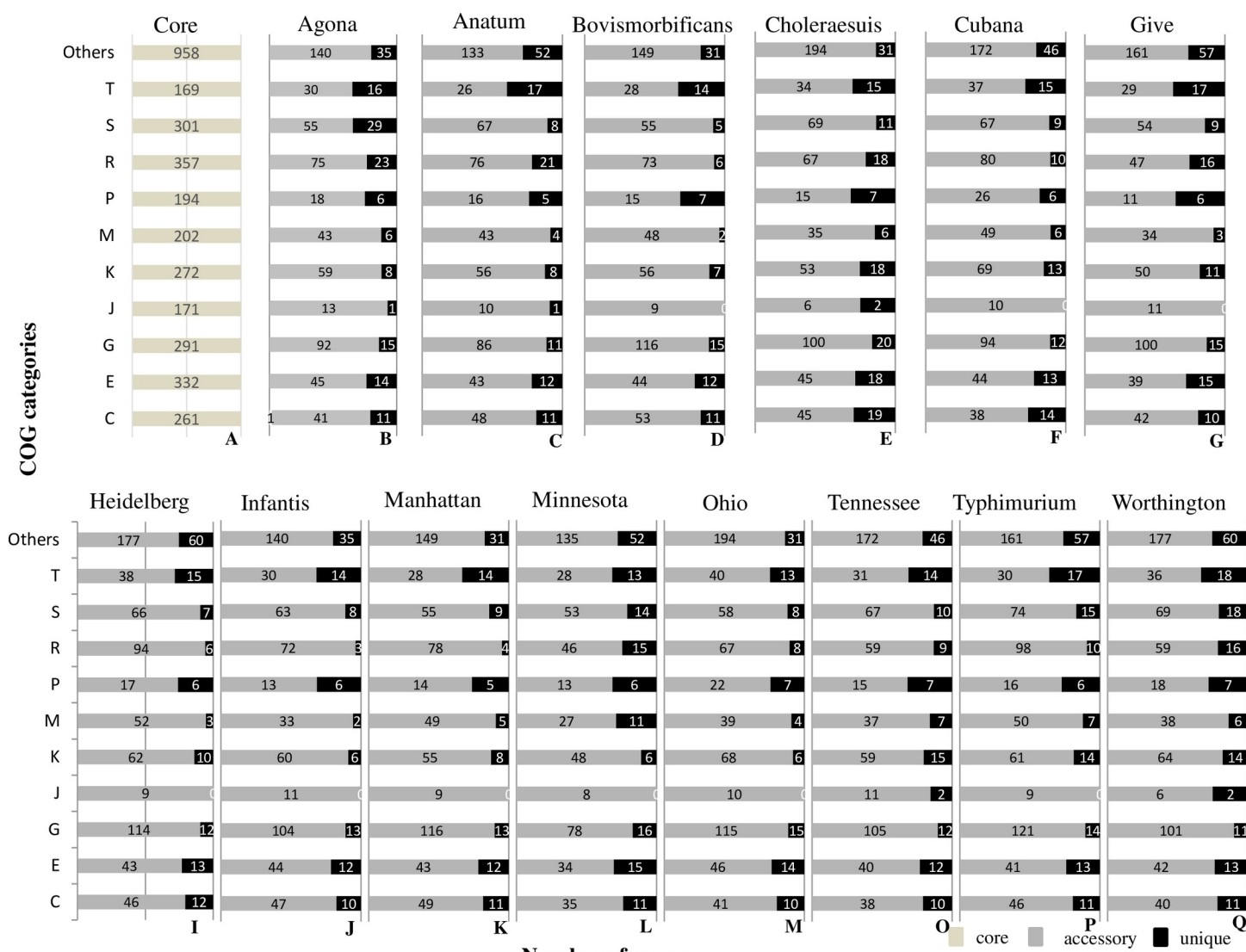

**Fig 4. Distribution of clusters of orthologous groups in core, accessory and unique genes in the genomes of different *Salmonella* serovars.** Common numbers of core genes in each COG categories were observed and presented in Fig 4(A), The bars represents the number of clusters of orthologous groups assigned genes present in accessory (grey bars) and in the unique genes (black bars).

Homologs of the *safA* (saf-pilin pilus formation protein) was identified in 6 of the 14 *Salmonella* serovars (Anatum, Bovismorbificans, Cubana, Give, Johanesberg and Manhattan), but these homologs were so diverse that they were included in the list of unique genes. Highly diverse toxin-antitoxin (TA) systems type II genes were detected in different *Salmonella* serovars, including *vapB* (Agona and Typhimurium), *higA* (Choleraesuis and Cubana), *yfjZ* (Minnesota and Manhattan) and *cbtA* (Bovismorbificans and Tennessee). UDP-L-Ara4N formyltransferase (*arnA*), a bifunctional genes facilitating polymyxin resistance, was detected in *S.* Choleraesuis. Highly diverse colonization factor antigen I subunit E (*cfaE*) gene was detected in *Salmonella* serovar Choleraesuis, Give, Minnesota and Worthington. Microcin-M immunity protein (*cmi*) was detected in *S.* Heidelberg.

The PHASTER analysis identified intact, questionable, and incomplete phage in genomes of the *Salmonella* isolates. Only intact phage were analyzed in this study and a total of 52 intact

**Table 4. Phage harboring virulence and resistance genes in *Salmonella* serovars.**

| *S. enterica* serovar | Isolate | No of Phage | Best match (kb) | Antibiotic resistance and virulence genes on Phage |
|---|---|---|---|---|
| **Agona** | CRJJGF_0019 | 4 | Aeromo_phiO18P_NC_009542 (18.4), Entero_fiAA91_ss_NC_022750 (28.6), Salmon_vB_SosS_Oslo_NC_018279(49.9), Entero_lato_NC_001422 (11) | – |
| **Anatum** | CRJJGF_0121 | 4 | Salmon_Fels_2_NC_010463(34.2), Salmon_SPN3UB_NC_019545(46.4), Haemop_HP1_NC_001697(26), Salmon_118970_sal3_NC_031940(63.3) | *sipABCD, invABCEFGHIJ, spaOPQRS, prgHIJK, sptP, sicAP, sopD,pipB2, orgBC, sicA,mig-14,* **mgtC** *tetC* |
| **Bovismorbificans** | CRJJGF_0070 | 3 | Shigel_SfII_NC_021857 (39.8), Gifsy_1_NC_010392 (55.1), Entero_lato_NC_001422 (5.6) | *sodC1* |
| **Choleraesuis** | CRJJGF_0148 | 4 | Salmon_g341c_NC_013059(37.1), Salmon_118970_sal3_NC_031940(36.5), Gifsy_2_NC_010393 (33.3), Phage_Gifsy_1_NC_010392 (23.1) | *sodC1* |
| **Cubana** | CRJJGF_0088 | 3 | Salmon_SEN34_NC_028699 (42.2), Entero_PsP3_NC_005340 (35.2), Entero_lato_NC_001422(66.3) | *SopB tetA* |
| **Give** | CRJJGF_0073 | 4 | Salmon_SEN34_NC_028699 (29.3), Entero_mEp235_NC_019708(26.3), Pseudo_PppW_3_NC_023006(26.3), Entero_lato_NC_001422(5.3) | *pipB* |
| **Heidelberg** | CRJJGF_0002 | 4 | Gifsy_2_NC_010393 (39.9), Entero_fiAA91_ss_NC_022750(33.9), Entero_lato_NC_001422(13.8), Entero_P4_NC_001609 (14.1), Salmon_118970_sal4_NC030919 (40.3) | *sodC1, grvA,* **gtrA** *bla*<sub>TEM-1</sub> |
| **Infantis** | CRJJGF_0031 | 3 | Entero_SfI_NC_027339 (41.6), Salmon_vB_SosS_Oslo_NC_018279(24.3), Salmon_vB_SosS_Oslo_NC_018279(39.6) | – |
| **Minnesota** | CRJJGF_0078 | 6 | Entero_I2_2_NC_001332 (5.2), Salmon_Fels_1_NC_010391 (14.7), Haemop_HP1_NC_001697(43.2), Aeromo_phiO18P_NC_009542 (32.8), Salmon_118970_sal3_NC_031940(44.1), Entero_lato_NC_001422 (10.7) | – |
| **Manhattan** | CRJJGF_0112 | 3 | Salmon_g341c_NC_013059 ((41.8), Entero_186_NC_001317 (37.5), Entero_lato_NC_001422(19.9) | – |
| **Ohio** | CRJJGF_0161 | 2 | Salmon_ST64T_NC_004348(38.6), Entero_lato_NC_001422(24.8) | *bla*<sub>TEM-1</sub> |
| **Tennessee** | CRJJGF_0089 | 3 | Salmon_vB_SosS_Oslo_NC_018279 (50.8), Haemop_HP1_NC_001697 (27.2), Entero_lato_NC_001422(13.7) | *aadA2, ere(A), bla*<sub>CARB-2</sub> |
| **Typhimurium** | CRJJGF_0051 | 6 | Entero_ST104_NC_005841 (42.7), Gifsy_2_NC_010393 (27.8), Salmon_118970_sal3_NC_031940(47.8), Salmon_118970_sal3_NC_031940 (75.1), Gifsy_1_NC_010392 (18.3), Entero_lato_NC_001422 (7.1) | *gyrA, sodC1, sspH2,* **rck, gtrA** *aadA2, bla*<sub>CARB-2</sub>*, sul1* |
| **Worthington** | CRJJGF_0141 | 3 | Gifsy_1_NC_010392 (31.8), Salmon_SPN3UB_NC_019545(47.2), Entero_lato_NC_001422(23.5) | *sspH1* |

*The virulence genes highlighted in bold are detected in resistance gene carrying phage

lysogenic phage were predicted by PHASTER and they are listed in Table 4. Most phage were highly diverse. The most common families included relatives of Gifsy-1 (NC_010392), Gifsy-2 (NC_010393), and Salmon_SEN34 (NC_028699). Gifsy-1 like phage were detected in *Salmonella* serovars Bovismorbificans, Infantis, Typhimurium, and Worthington, Gifsy-2 like phage were detected in *Salmonella* serovars Choleraesuis, Heidelberg and Typhimurium, and the Salmon_SEN34 like phage were detected in *Salmonella* serovars Choleraesuis, Cubana and Give. Individual phage found in only one of the 14 included in Cubana (Entero_Psp3 [NC_005340]), Heidelberg (Escher_D108 [NC_013594]; Entero_P4 [NC_001609]), Minnesota (Entero_I2_2 [NC_001332]; Entero_SfV [NC_003444]; Salmon_vB_SemP_Emek [NC_018275]), Manhattan (Salmon_epsilon34 [NC_011976]; Entero_186 [NC_001317]) and Typhimurium (Entero_ST104 [NC_005841]). Of particular interest is the fact that some of the phage carried non-phage "cargo" genes. The AR beta-lactamase gene blaTEM-1 was seen in S. Heidelberg (Salmon_118970_sal4_NC030919), S. Ohio (Entero_lato_NC001422). The blaCARB-2 gene was seen in S. Tennessee (Entero_lato_NC001422) and *S*. Typhimurium (Salmon_119870_sal3_NC031940). The tetracycline gene *tetC* was seen in *S*. Anatum (Salmon_118970_sal3_NC031940);

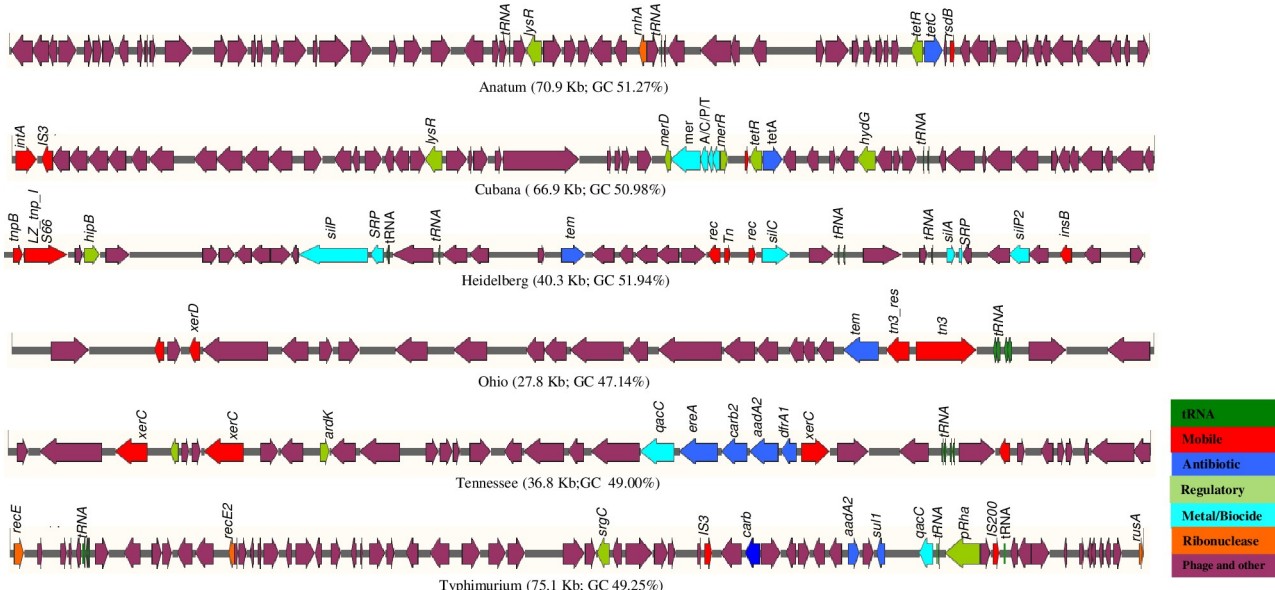

**Fig 5. Examples of integrated bacteriophage.** Cargo genes are annotated, such as antibiotic and other resistance genes (*Hg* resistance gene [*mer*A/C/P/T/ and mer regulators *mer*D/R; *S*. Cubana], *Ag* resistance genes [*sil*A/C/P/P2, SRP; *S*. Heidelberg], quaternary ammonium compound gene [*qac*C; *S*. Tennessee and *S*. Typhimurium]). The arrow shows orientation of the genes in the contigs and genes are color coded to define different gene categories.

and *tetA* was seen in *S*. Cubana (Entero_lato_NC001422). The aminoglycoside gene *aadA2* was seen in S. Tennessee (Entero_lato_NC001422) and *S*. Typhimurium (Salmon_118970_-sal3_NC031940). Our analysis further confirmed that blaCARB-2 was harbored on the class 1 integrons In1582 and In167 that integrated into the possible phage Entero_lato_NC001422 (S. Tennessee) and Salmon_119870_sal3_NC031940 (S. Typhimurium), respectively. We also observed that metal resistance genes *merDACPTR* and *silA/C/P/P2* were on the possible prophage Entero_lato_NC_001422 and Salmon_118970_sal4_NC030919 of *S*. Cubana and *S*. Heidelberg, respectively (Fig 5).

These phage were further analyzed for the presence of virulence genes and results are summarized in Table 4. Eight of fourteen isolates harbored at least one virulence gene on phage. The comparisons for the homologous regions in prophage revealed 10 different phage shared similar regions, although these prophage were not 100% identical, but showed similarity ranging between 40 to 98%. The linear comparisons of the phage are presented in Fig 6. The identical regions were mostly part of the phage related genes.

CRISPR loci were identified in the majority of isolates and these loci were compared for homology across serovars. The common spacers were observed towards the ancestral end of the CRISPR array *e.g.* common spacer was identified in eight serovars (Agona, Cubana, Choleraesuis, Heidelberg, Infantis, Manhattan, Typhimurium and Worthington). Spacer numbers varied between serovars and across the CRISPR loci (Table 5).

The sequential identical spacers towards the ancestral end was identified among these serovars for *eg.* two spacers detected in *Salmonella* serovar Cubana and Worthington, three spacers detected in *Salmonella* serovar Agona, Heidelberg and Typhimurium, four spacers detected in *Salmonella* serovar Heidelberg and Typhimurium. Unique spacer arrays were observed in serovar Minnesota, Ohio and Tennessee (the last spacer i.e. spacer 57 was identical to spacer 2 of serovar Anatum). The signature protein of the Type I CRISPR systems *Cas3* gene was detected in all the *S*. serovars and variations were noticed in this gene when compared with *S*. typhimurium

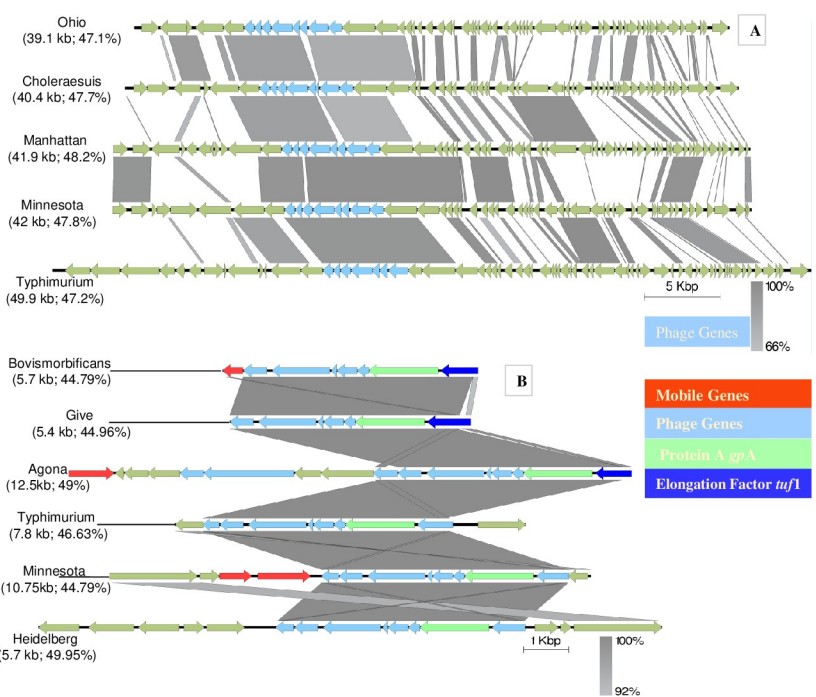

**Fig 6. Alignment of closely homologous intact integrated phage among *Salmonella* serotypes.** A. Phage genes synteny in five serotypes. B. Phage gene arrangement along with mobile genes (transposase, recombinase) in six serotypes. The arrow shows orientation of the genes and genes are color coded to define mobile, phage and other genes.

LT2 *cas3* as summarized in Table 5. Spacer sequences were also analyzed to detect target prophage with the help of Blast using UniProt phage sequences (https://www.uniprot.org/). Prophage targets were observed in most CRISPR loci, with the most common prophage targets being tail, head, and capsid proteins. Besides this, the other targets were integrase and recombinase regions.

## Discussions

The implementation of WGS allows broader inference of pathogen characteristics including prediction of antibiotic resistance and virulence profiles from the sequences. WGS has been previously used for the prediction of AR genes in a wide range of microbes including *Salmonella* [38]. Phenotypic and genotypic correlation analysis confirmed high concordance (97%) between phenotypically confirmed and *in silico* predicted AR genes. No absolute concordance was observed due to insertional inactivation if *tet(B)* gene in Choleraesuis and no resistance phenotype was shown by this isolate (Table 2). Resistance to aminoglycosides is either due to inactivation or modification by acetyltransferases, phosphotransferases, and nucleotidyltransferases [39]. The aac6-I variants Y and aa of was detected in all the isolates and located on the chromosome. This cryptic gene has previously been reported in *Salmonella* that was due to a deletion in the promoter region [40]. The s*trA/B* and *aadA* genes are frequently associated with MGE that easily disseminate aminoglycoside resistance genes in *Salmonella* and other gram negative bacteria; streptomycin resistance has also been used as an important epidemiological marker to indicate the possibility of MDR in pathogens [41–43]. Many class 1 integrons harbor *aadA* gene variants [44]; four *aadA* variants were associated with class 1 integrons in this analysis. Other aminoglycoside resistance genes that confer resistance to kanamycin (*aph* (3")-Ia) and hygromycin B (*aph*(4')-Ia) were predicted in this analysis and these genes have

**Table 5. Prophage target identified in spacer sequences in CRISPR arrays in *Salmonella* serovars.**

| *Salmonella* serovar | % identity with cas3 gene | Number of CRISPR array | Spacer | | |
|---|---|---|---|---|---|
| | | | No | Target | Gene |
| **Agona** | 98.20 | 21 | 8 | Prophage | Phage *EaA* protein |
| | | | 10 | Prophage | Portal protein |
| **Anatum** | 98.87 | 24 | 5 | Prophage | Tail tube |
| | | | 8 | Prophage | Phage integrase |
| | | | 11 | Prophage | Phage *EaA* protein |
| | | | 15 | Prophage | Phage-related protein |
| **Bovismorbificans** | 45.42 | 23 | 18 | Bacteria/Prophage | Integrase |
| | | 17 | 14 | Bacteria/Prophage | Repressor |
| **Cubana** | 99.32 | 21 | 6 | Prophage | Bacteriophage Mu |
| | | | 14 | N/A | Uncharacterized protein |
| | | 9 | 6 | Prophage | Adenine methylase |
| **Give** | 45.62 | 15 | 4 | Prophage | Tail fiber protein |
| | | | 6 | Prophage P1 | DNA replication |
| | | | 8 | Prophage | Integrase |
| | | 18 | 14 | Prophage lambda | *ninG* |
| **Heidelberg** | 99.32 | 18 | 8 | Prophage | Phage-related protein |
| **Infantis** | 99.21 | 26 | 14 | Prophage | Repressor protein |
| | | 30 | 7 | Prophage | Phage protein |
| | | | 11 | Caudovirus | Capsid protein |
| | | | 13 | Prophage | Uncharacterized protein |
| | | | 18 | Prophage | DNA-binding protein |
| **Manhattan** | 98.42 | 11 | 8 | Prophage | Phage EaA protein |
| | | | 10 | Prophage | Portal protein |
| **Minnesota** | 46.74 | 15 | 3 | Prophage | Tail proteins |
| | | | 10 | Spiroplasma phage | Protein |
| | | | 12 | *Escherichia* phage RCS47 | Head protein |
| **Ohio** | 99.21 | 24 | 15 | Prophage lambda | Terminase large subunit (*GpA*) |
| **Tennessee** | 45.45 | 22 | 20 | *Salmonella* phage SE1 | Uncharacterized protein |
| | | | 22 | *Salmonella* phage SEN34 | DNA polymerase III theta subunit |
| | | 57 | 17 | Prophage | Terminase |
| **Typhimurium** | 99.89 | 26 | 8 | Bacteria/Prophage | Tyrosine recombinase *xerC2* |
| | | | 21 | Prophage | Uncharacterized protein |
| **Worthington** | 97.97 | 28 | 1 | Prophage | Integrase |
| | | | 20 | Prophage | Terminase |
| | | | 23 | Prophage | Tail |
| | | | 26 | Bacteria | Chromosome partitioning protein *parB* |
| | | 16 | 9 | Prophage | Phage protein |
| | | | 14 | Prophage | Regulatory Protein |

also been previously identified in WGS data of *Salmonella*; moreover, *aph*(*3"*)-*Ia* was the most frequently detected gene from food sources [19]. All three β-lactam resistance genes (*bla*$_{\text{TEM-1}}$, *bla*$_{\text{CMY-2}}$ and *bla*$_{\text{CARB-2}}$) are highly prevalent in *Salmonella* isolated from U.S. animals, and humans. These genes are the mostly horizontally acquired β-lactamases in *Salmonella* [42, 45]. Tetracycline is widely used in food animals to combat respiratory infections and have also been used as growth enhancer and an additive in feed [46](45,46). The genes conferring resistance to tetracycline are widespread among *Salmonella* serovars and easily transferred among

wide range of microbes through HGT [47, 48]. All isolates except one with *tet* genes were resistant to tetracycline in this analysis and only tetracycline efflux pump genes were identified. No other *tet* genes (ribosomal protection protein or inactivated enzyme) were detected which suggested that the efflux pump mediated resistance was the major tetracycline resistance mechanism among these *Salmonella* serovars. The high prevalence of these efflux pump genes have been reported in *Salmonella* species [49] which further confirmed our findings (Table 2). The erythromycin resistance gene *ere*(A) was detected on a class 1 integrons that harbored *dfrA*, *bla*CARB-2 and *aadA2* resistance determinants (Table 3). The *ere*(A) gene has previously been reported in class 1 integrons [50]. All *dfr* variants (A1, A7, A12 and A16), are associated with different profiles of class 1 integrons encoding resistance to trimethoprim. The class 1 integron carrying *dfrA* gene variants and other resistance genes have been reported from several *Salmonella* serovars Anatum, Choleraesuis, Corvallis, Eppendorf, Gallinarum, Kentucky, Rissen, Stanley, Schwarzenrund, Typhimurium, and Weltevreden [51–54]. Identical class 1 integron gene cassette (*In363*) were detected in *Salmonella* serovar Agona and Ohio (Table 3), that indicated that inter or intra species transfer of integrons help in the spread of antimicrobial resistance genes among bacteria. The fosfomycin resistance *fosA* gene alleles are located on the chromosome.The *floR* genes has been isolated from a wide range of animals and humans and it has been previously reported from various *Salmonella* serovars including *S.* Typhimurium. Chloramphenicol was used to treat MDR infections in human and animals [55–57]. The sulfisoxazole resistance genes *sul1* (located within 3'-conserved segment [3'-CS] of class 1 integrons) and *sul2* (associated with small multicopy plasmids or large transmissible multiresistance plasmids) are the most frequently found genes for sulfonamide resistance among sulfonamide-resistant isolates from food animals and humans, whereas *sul3* is detected in various large *Salmonella* plasmids [58, 59]. The sulfonamide resistance genes *sul1* and *sul2* were the most abundant AR genes detected in this analysis, which was in accordance with the previous study where high prevalence of these genes has been reported in the MDR *Salmonella* isolated from animals, humans and retail meats from Northern America [60]. A secondary mutation in *parC* without primary mutation in *gyrA* is ineffective to make these isolates resistant to nalidixic acid; however, these isolates may become highly resistant once primary mutation occurs in *gyrA* gene as reported previously where several fold increase in resistance was observed when a secondary mutation in *par*C was seen in an isolate that had a primary mutation in *gyr*A [61, 62].

The AR genes carrying contigs were distinct and none of them were identical among *Salmonella* serovars except AR contigs of *S.* Agona and *S.* Ohio that shared 99.9% sequence similarity and gave best hit with virulence-resistance plasmids (*incA/C2*) of *S.* Typhimurium [63]. This >35.0 kb fragment carried *tet*A, *tet*R, *str*A, *str*B, *sul2* and *bla*CMY-2. The *str*A, *str*B, *sul2* AR genes was flanked by Tn7 elements in this partial *incA/C2* plasmid and this was the most frequently recorded resistance gene combination among *Salmonella* serovars located on small broad-host-range plasmids, as well as detected on the chromosome in *Salmonella* [49]. Another 5.0 kb fragment from *S.* Bovismorbificans carrying *str*A and *str*B best matched with the genome of *S.* Enteritidis (CP020442 and CP022069) and showed 99% coverage and 100% identity to the region and appear to be widely distributed in *Salmonella* and other gram-negative bacteria [49]. These genes have been described as being part of transposon Tn*5393* and have also been identified in bacteria circulating in humans, animals, and plants [64]. Another contig carrying *aad*A, *ebr* and *sul1* was detected in *S.* Cubana and gave best hit with *E. coli* plasmid pBM0133 (KJ170699)/ pDGO101 and this array of genes is harbored by class 1 integron *In1581* (Table 3) [65, 66]. The contig from *S.* Heidelberg that harbored neomycin (*aph*A2), bleomycin and streptomycin (*sph*) resistance genes best matched with the *E. coli* transposons Tn5 (U00004). These genes (*aph*A2 and *sph*) are encoded by Tn5 and used as selectable

markers in cloning vectors for both eukaryotes and prokaryotes [67, 68]. A 8.69 kb contig from *S*. Choleraesuis carrying *str*A, *str*B and *sul*2 gave best hit with TY474p3 plasmid (CP002490) and it was a closed plasmid [49]. Another 5.5Kb contig carrying *ere*A, *aad*A2, *bla*-*CARB-2* and *dfrA16* from *S*. Tennessee gave best hit with *E. coli* integron *int*l1 (KX57988). This integron carrying *ere*A and *aad*A2 is rarely reported [69], while a 5.0Kb fragment carrying *floR* and *tetA* in *S*. Typhimurium, this fragments was the part of *Salmonella* genomic island 1 (SGI1) that best matched with the regions of complete genome of *Salmonella* Typhimurium (CP028318:4845261–4850261, and CP014979:42459–47459). SGI1 has been identified in several *Salmonella* serovars including Typhimurium, Agona, Paratyphi B, Albany, Meleagridis and Newport [70–74]. The detection of SGI1 in different *Salmonella* serovars from various sources including animals and humans across the world indicates frequent dissemination of the SGI1 through horizontal transfer [75].

The size of the core genome in this analysis was in accordance with the previous studies where the size of the pan-genome expanded slightly while the size of core genome shrunken with addition of new genomes. The inter serovars core genome size (3, 224 core genes) within the 35 *Salmonella* subsp. *enterica* was lower while the intra serovar core genome size (3, 836 core genes) for *S*. Typhimurium was higher compared to our analysis [76, 77]. Low number of unique genes was directly related to the reduced genomes size, where HGT events were less and *S*. Bovismorbificans well correlated in this context with lowest numbers of unique genes and lower genome size. We could not draw any correlation between high number of unique genes and with large genome size. However, moderate genome length with high unique genes revealed its evolutionary significance. Uniform distribution of genes was noticed for other COG categories among *S*. serotypes. The abundance of E, G, and K cog categories of genes increases the diversity, uniqueness and their complex transcriptional regulatory networks that support morphological and physiological differentiation. The abundance of G and L functional categories of genes in dispensable genes clearly showed that these functional genes acquired for the better adaptability and diversity [78].

The majority of virulence genes were part of the core genes and essential for pathogenicity and infection. WGS data helped to correctly interpret the variations in genes among *Salmonella* serovars for better understanding of pathogen and evolution. This analysis clearly identified some of the genes specifically detected in one or several *Salmonella* serovars. For example, the *pef* operon comprising *pefABCD* genes that is needed to form structural fimbria and mediate the binding of bacteria to the microvilli of enterocytes and *spvBCR* that enhance virulence were solely identified in *S*. Typhimurium; these operons were not detected in *S*. Choleraesuis. However, they have been reported in virulence plasmid pSCV50 [79] harbored in *S*. Choleraesuis. *RatB* which encodes a secreted protein associated with intestinal colonization and persistence was detected in all the analyzed serovars except Agona, Anatum, Cubana, Tennessee, and Worthington [80] while *gogB* an anti-inflammatory effector that limits tissue damage during infection was detected in serovars Typhimurium and Choleraesuis which are predominantly associated with swine [81, 82]. Several studies have demonstrated the loss of genes associated with host specificity, thus some microbes only adapted to become a better pathogen to specific hosts while losing the ability to infect other potential hosts [83]. The *lpf* operon has been reported in *Salmonella* and is involved in the adhesion to the small intestine [84]. This operon was detected in most *Salmonella* serovars with high variations in *lpfD* genes that are highly host specific; variations in these genes could be an important factor that may influence the host range. The conserved nature of *phoP/Q* genes suggests that its pivotal regulatory role in *Salmonella* promotes several phenotypes including cationic antimicrobial peptides resistance which enhance outer membrane barrier function, and the ability to cause invasive and systemic disease [85, 86].

The genes which are confined to one serovar or their highly variable variants detected among serovars were mostly acquired genes and many of them were harbored on phage or plasmid. The T6SS is a versatile secretion system widespread among Gram-negative bacteria and directly involved in a variety of cellular processes, including virulence in several bacterial pathogens [87, 88]. Homology search resulted in best hits with *Salmonella* serovars Weltevreden (LN890520), Agona (CP015024), and Sloterdijk (CP012349). These loci have been reported to encode T6SS harbored on SPI-19 in Agona and Weltevreden [89]. *safA* plays a critical role in host recognition; highly diverse *safA* genes in six different *Salmonella* serovars confirmed that these serovars could be the part of swine asymptomatic carriers. Whereas highly identical (>98%) *safA* was found in *S.* Choleraesuis and *S.* Typhimurium which may recognize similar hosts as these two serovars are predominantly associated with swine [90]. The *cfa* gene in *S.* Choleraesuis was highly conserved and were identical with other *S.* Choleraesuis in GenBank database (CP012344 and CP007639) and it may be essential for the attachment to a specific host [91]. The antimicrobial peptides encoding *cmi* from *S.* Heidelberg best matched with *Escherichia coli* plasmids harboring this gene (CP035337 and CP035355) and these plasmids harbored several AR genes. We believe *cmi* was transferred to *S.* Heidelberg as an accessory gene with other AR genes.

The integration of phage genetic elements in the genomes results in genome diversification of closely related bacterial strains [92] as these phage harbors AR and virulence factor that can enhance bacterial fitness [93]. Phage-associated resistance and virulence genes have been reported in *Salmonella* revealing the role of phage-transferred genes in pathogenicity and resistance [94, 95]. Gifsy-1 and Gifsy-2 like phage were highly prevalent among *Salmonella* serovars and these phage have previously been reported in *Salmonella* [96, 97]. In this study, we found a number of phage that encode for resistance and virulence genes. For example, we identified a 75.1 kb phage that harbor class 1 integrons carrying *aadA2*, *blaCARB-2*, *sul1* AR genes and encodes a glucosyl transferases *gtrA* in the serovar Typhimurium, this putative virulence gene has previously been reported in P22-like and P2-like prophage in different *Salmonella* serovars [98].

The analysis and interpretation of the CRISPR region is complex. Spacers can only be added to the 5′ end and deleted anywhere in the locus, leaving the 3′-spacer as ancestral. The conserved spacers at the ancestral ends allowed us to understand the common infection history or common ancestry of these serovars though there is a possibility of the degradation of many internal spacers [17]. A careful analysis of CRISPR locus alignments provided an evolutionary framework to view the phylogenetic patterns of CRISPR diversity as we observed deletion of spacers in serovars Choleraesuis, Heidelberg and Typhimurium. Prophage targets were observed in most CRISPR loci; and about 87% (33/38) of spacer sequences were found to match with sequences from phage suggesting that phage were an important type of HGT in *Salmonella*. In this study, all eight *cas* genes pattern was not identified in *S.* Worthington as draft genome sequences was used and missing sequencing data cannot be ruled out. Cas3 involved in the cleavage of invading DNA and considered as an important component of the CRISPR mechanism and it was the only *cas* gene detected in all the serovars [99]. Comparison of cas3 among the serovars confirmed two distinct pattern,that was in accordance with earlier report where two distinct cas3 pattern was reported [100].

Vehicles of dissemination (plasmid, phage, integrons *etc.*) are crucial for microbial evolution and they play a major role in genetic innovation and genome evolution. Resistance determinants were located either on plasmids, integrons or integrons were also associated with phage which indicated that they were acquired horizontally. Highly diverse class 1 integrons containing different AR gene cassettes were detected among *Salmonella* serovars and these AR genes accounted for resistance to relative drugs. This suggests that these determinants of AR

are functional and crucial for the survival of the microbes. A fraction of the virulence determinants were also identified on mobile elements and this suggests that these determinant are crucial for better survival. The common ancestral ends in CRISPR suggest that these isolates shared common ancestry and microbes are continuously acquiring new phage to counter the CRISPR regulation.

## Conclusions

The insight into the WGS data enabled us to virtually access the genetic content for the assessment of genetic diversity, pathogenesis, evolution, serotyping, virulence and resistance profiling that enhanced our understanding of the genomic diversity among *Salmonella* serovars. The pangenome size was in accordance with the earlier report however the core genome was comparatively higher due to less isolates used in this analysis. Most of the virulence factors were part of the core genomes indicating absolutely required factor for virulence; however, a small fraction of virulence factors were highly diverse (*safA*) or confined to one isolates (*cfa*, *pefABCD*, *and spvBCF*) indicating variation in pathogenicity among *Salmonella* serovars. Inter and the intra species transferability of MGE enhance their reachability to the specific or the broader host. Variable MGE information from the WGS enables understanding of the dynamics of horizontal gene transfer carrying resistomes that were widely distributed in other Gram-negative bacteria. Detection of class 1 integrons carrying resistance determinants (*In1582* and *In167*), high sequence diversity among related prophage carrying various resistance and virulence genes, suggesting extensive horizontal gene transfer that enhanced bacterial fitness. WGS enabled us to obtain a comprehensive resistome profile that can be useful to develop antibiotic resistance combat strategies. Comprehensive genome analysis with large dataset could profoundly enhance our understanding of the adaptability and survival mechanism in *Salmonella* serovars.

## Supporting information

**S1 Table. Virulence profiles of *Salmonella* serovars.**
(XLSX)

**S2 Table. Analysis of mutations in Chromosomal genes conferring resistance to antibiotics in Salmonella serovars.**
(XLSX)

**S3 Table. Gene content in *Salmonella* serovars confined to one serovars.**
(XLSX)

**S1 Fig. Bar graph represents the distributions of the genes in core, accessory and unique gene in different Salmonella serovars from swine.**
(TIF)

## Acknowledgments

SKG thanks Calvin Williams for IT support at USNPRC facility. MM thanks Weiping Chu, Steffen Porwollik, and Prerak Desai, who were involved in genomic DNA assessment, sequence storage, and confirmation.

## Author Contributions

**Conceptualization:** Sushim K. Gupta, Charlene R. Jackson, Jonathan G. Frye, Michael McClelland.

**Data curation:** Sushim K. Gupta.

**Formal analysis:** Sushim K. Gupta, Poonam Sharma.

**Funding acquisition:** Charlene R. Jackson, Jonathan G. Frye, Michael McClelland.

**Investigation:** Sushim K. Gupta, Elizabeth A. McMillan.

**Methodology:** Lari M. Hiott, Tiffanie Woodley, John B. Barrett, Jonathan G. Frye.

**Supervision:** Jonathan G. Frye.

**Validation:** Poonam Sharma, Elizabeth A. McMillan, Charlene R. Jackson, Lari M. Hiott, Tiffanie Woodley.

**Writing – original draft:** Sushim K. Gupta.

**Writing – review & editing:** Sushim K. Gupta, Poonam Sharma, Elizabeth A. McMillan, Charlene R. Jackson, Lari M. Hiott, Tiffanie Woodley, Shaheen B. Humayoun, Jonathan G. Frye, Michael McClelland.

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
