## [Decision Letter · Decision Letter 0]

6 Sep 2019

PONE-D-19-17181

Genomic comparison of diverse Salmonella serovars isolated from swine

PLOS ONE

Dear Dr. Frye,

Thank you for submitting your manuscript to PLOS ONE. After careful consideration, we feel that it has merit but does not fully meet PLOS ONE’s publication criteria as it currently stands. Therefore, we invite you to submit a revised version of the manuscript that addresses the points raised during the review process.

Your manuscript has been reviewed by two experts. Based on their comments, a major revision is needed  before a decision can be made.

We would appreciate receiving your revised manuscript by 4 weeks. To enhance the reproducibility of your results, we recommend that if applicable you deposit your laboratory protocols in protocols.io, where a protocol can be assigned its own identifier (DOI) such that it can be cited independently in the future. For instructions see: http://journals.plos.org/plosone/s/submission-guidelines#loc-laboratory-protocols

We look forward to receiving your revised manuscript.

Kind regards,

Yung-Fu Chang

Academic Editor

PLOS ONE

Journal Requirements:

2. In your Financial Disclosure statement, please ensure you have stated whether the funders had any role in study design, data collection and analysis, decision to publish, or preparation of the manuscript.

Reviewers' comments:

Reviewer's Responses to Questions

**Comments to the Author**

1. Is the manuscript technically sound, and do the data support the conclusions?

Reviewer #1: Yes

Reviewer #2: Yes

2. Has the statistical analysis been performed appropriately and rigorously? 

Reviewer #1: N/A

Reviewer #2: Yes

3. Have the authors made all data underlying the findings in their manuscript fully available?

Reviewer #1: Yes

Reviewer #2: Yes

4. Is the manuscript presented in an intelligible fashion and written in standard English?

Reviewer #1: Yes

Reviewer #2: Yes

5. Review Comments to the Author

Reviewer #1: Gupta et al., examined the genomic profiles of diverse Salmonella serovars from swine using whole genome sequencing. Overall there are some interesting pieces, but the paper needs some major revisions and editing. Many of the specific areas of concern or further consideration are provided below.

Title

Ln3: Please remove additional ‘+’ symbol after the author name Jonathan G. Frye.

Abstract

Ln44: “More than 75% of the genes were part of the core genome”, according to Ln252 it is ~ 42% please confirm.

Ln52: Expand CRISPR.

Ln55: The keywords Salmonella and serovar can be removed and add new keywords because these two are appearing in title and arrange the keywords in descending order.

Introduction

Ln73: Please remove the” >” symbol because” more than” is already written.

Ln76-78: In general objective and hypothesis are written at the end of the introduction. However, they are written at Ln76-78.

Ln78: were Salmonella serovars collected from swine swabs?

Materials and Methods

Ln108-110: what are concentrations used for antibiotics panel used? Which solvent did used for solubility?

Results

Ln143: “This study was the part of retrospective study” were the samples used in this study obtained from the McMillan et al., 2019 (ref # 22)? It is not clear in M&M section.

Ln152-157: Data not available in Table 1.

Ln156: Expand ‘ANI’.

LN158: Expand ‘SRA’.

Ln164: Expand ‘CLSI’.

Ln172: Please correct the sentence.

Ln177-179: The information related to ‘total number of AR genes’ is not available in Table 1.

Ln178: Please include the total number of genomes before ‘genome sequences’.

Ln179-Ln182: Please recheck the result, the gene variants Y and aa of aac6-I were not found in the table 2.

Ln184: According to table2, only seven isolates were resistance to beta-lactams. Please check.

Ln198: blaCARB-2 is 2/14 right?

Ln207-209: Please check the results, the gene “ereA” is not found in Agona. Please restructure the sentence.

Ln211: “both “only in Agona both sul1 and sul2 were detected but not in typhimurium were only sul1 were detected.

Ln213: Please remove ‘chloramphenicol resistant’ or rearrange the sentence because not all the fourteen are resistance to chloramphenicol.

Ln214-215: What is the need for writing cml or cat resistance genes here because they are not detected in the study.

Ln215-220: Data not available.

Ln221: How were integrons detected? This information is not available in M & M section.

Ln231: How were plasmids detected? This information is not available in M & M section.

Ln233-237: The replicon types such as IncII, IncFII, IncN, IncP6, col440II and IncQ were not found in table 2.

Ln240-241: Please provide the complete legend for Fig.1 Example what does arrows indicated?

Ln252: The total number of unique genes 2,324 is correct? From Fig.2 they are 2,517. Please make changes.

Ln255: Please include how floral diagram is created in the M & M section?

Ln265: Please remove’(ref)’

Ln266: I could see only 22 COG functions in figure 3.

Ln269-270: Suggestions to change Y-axis as percentages and keep only most abundant functions (may be top 10 + combining all other functions).

Ln273: Please include number of genes near “highest and lowest”.

Ln275: Expand ‘E’ completely.

Ln288: Please include exact percentage values for G and L.

Ln293-328: These results are based on supplementary tables and these are presented for each isolate and it is difficult to compare the results. A comprehensive information like table 2 will help to look over the results clearly.

Ln357-359: Fig.5 results were not discussed in results section.

Ln362: This is ten right?

Discussion

Discussion is well organized however it is too lengthy, restatement of data (restatement of results) observed in some places.

References

Too many references.

Figures

Except figure 2, none of the figures are in good quality they are not clearly visible for interpretation, in addition more information is needed to add in the legend for easy interpretation.

Reviewer #2: The manuscript reports an extensive genome analysis of 14 diverse Salmonella serotypes isolated from swine using whole genome sequencing. The results suggest that majority of the genes were part of core genome and dispensable genes were acquired for adaptability. Here, the authors have performed analysis using various bioinformatics tools to identify and predict genetic repertoire in salmonella serovars. I found this paper of broad relevance to the field, especially to antibiotic resistance and public health. However, I have provided general comments and suggestions below that are necessary to improve the manuscript.

Minor comments

1. The authors have stated high concordance between phenotypically confirmed resistances and detected ARGs, however they did not state why it was not well correlated. The authors need to explain.

2. The authors noticed two distinct pattern of CRISPR-cas3 gene, how did they validated the findings ?

3. The authors have showed clusters of orthologous genes in core and accessory genes (figure 4). Authors might also consider to show the distribution of cogs in unique genes.

4. Authors might consider to add a figure showing the distribution of pan genome (core genes, accessary genes and unique genes) content in each salmonella serotypes (for eg: serovars in x-axis and number of genes in y-axis).

5. The authors need to elaborate the materials and methods sections.

6. Authors should add cogs database reference (line 265)

6. PLOS authors have the option to publish the peer review history of their article (what does this mean?). If published, this will include your full peer review and any attached files.

Reviewer #1: Yes: Nagaraju Indugu

Reviewer #2: No

---

## [Author Response · Author response to Decision Letter 0]

4 Oct 2019

Reviewer #1:

Gupta et al., examined the genomic profiles of diverse Salmonella serovars from swine using whole genome sequencing. Overall there are some interesting pieces, but the paper needs some major revisions and editing. Many of the specific areas of concern or further consideration are provided below.

Title

Ln3: Please remove additional ‘+’ symbol after the author name Jonathan G. Frye.

Response: We have deleted the additional + symbol in Ln3 in the revised manuscript.

Abstract

Ln44: “More than 75% of the genes were part of the core genome”, according to Ln252 it is ~ 42% please confirm. 

Response: We would like to thank the reviewer for this observation, the authors want to convey that >75% of the genes in each genome were part of the core genome. To avoid confusion we have corrected the text accordingly in Ln 44. 

Ln52: Expand CRISPR. 

Response: CRISPR was expanded in Ln 52.

Ln55: The keywords Salmonella and serovar can be removed and add new keywords because these two are appearing in title and arrange the keywords in descending order.

Response: New key words have been added and arranged in descending order in Ln 57 in the revised manuscript.

Introduction 

Ln73: Please remove the” >” symbol because” more than” is already written.

Response: “>” symbol has been deleted in Ln71 in the revised manuscript.

Ln76-78: In general objective and hypothesis are written at the end of the introduction. However, they are written at Ln76-78.

Response: We would like to thank the reviewer for this observation. The text has been deleted and added at the end of the introduction (Ln 92-97). 

Ln78: were Salmonella serovars collected from swine swabs?

Response: The Salmonella serovars were from swine swabs.

Materials and Methods 

Ln108-110: what are concentrations used for antibiotics panel used? Which solvent did used for solubility? 

Response: All the antibiotic concentrations were as per Clinical and Laboratory Standards Institute (CLSI) recommendations. Minimum inhibitory concentrations (MICs) were manually recorded by SensiTouch®, and CLSI standards were used to determine resistance. The antibiotics were solublized in adequate solvent as per manufacturer’s recommendation. 

Results

Ln143: “This study was the part of retrospective study” were the samples used in this study obtained from the McMillan et al., 2019 (ref # 22)? It is not clear in M&M section.

Response: This is a comprehensive study in which different Salmonella serovars genomes exclusively from swine were analyzed. The isolates were obtained from the previous study.

Ln152-157: Data not available in Table 1.

Response: Adequate data has been added in the Table 1 in the revised manuscript.

Ln156: Expand ‘ANI’. 

Response: ANI has been expanded in the text in Ln155 in the revised manuscript.

LN158: Expand ‘SRA’.

Response: SRA has been expanded in the text in Ln157 in the revised manuscript.

Ln164: Expand ‘CLSI’.

Response: CLSI has been expanded in the text in Ln164-165 in the revised manuscript.

Ln172: Please correct the sentence.

Response: The sentence has been corrected in Ln172-173 in the revised manuscript.

Ln177-179: The information related to ‘total number of AR genes’ is not available in Table 1.

Response: The information regarding total number of AR genes has been added in Table 1.

Ln178: Please include the total number of genomes before ‘genome sequences’.

Response: We are not clear what the Reviewer is asking here. We sequenced 194 genomes from different food animals out of which 14 genomes from swine were used in this study.

Ln179-Ln182: Please recheck the result, the gene variants Y and aa of aac6-I were not found in the table 2. 

Response: The chromosomally located cryptic gene aac6-IY and aac6-Iaa were uniformly detected in all the isolates. It does not confer any resistance due to mutation in the promoter region, thus these genes were not included in Table 2. 

Ln184: According to table2, only seven isolates were resistance to beta-lactams. Please check.

Response: We would like to thank the reviewer for this observation. The text has been corrected in Ln184-185 in the revised manuscript.

Ln198: blaCARB-2 is 2/14 right?

Response: We would like to thank the reviewer for this comment. The digits has been corrected in Ln199 in the revised manuscript.

Ln207-209: Please check the results, the gene “ereA” is not found in Agona. Please restructure the sentence.

Response: We would like to thank the reviewer for this comment. The sentence has been restructured in Ln208-209 in the revised manuscript.

Ln211: “both “only in Agona both sul1 and sul2 were detected but not in typhimurium were only sul1 were detected.

Response: We would like to thank the reviewer for this comment. We have reframed and included the serovars details in Ln210-212 in the revised manuscript.

Ln213: Please remove ‘chloramphenicol resistant’ or rearrange the sentence because not all the fourteen are resistance to chloramphenicol. 

Response: We would like to thank the reviewer for this comment. We have reframed the sentence in Ln214 in the revised manuscript.

Ln214-215: What is the need for writing cml or cat resistance genes here because they are not detected in the study. 

Response: We have deleted cml or cat resistance genes.

Ln215-220: Data not available. 

Response: The data has been added as a supplementary table (S2 Table).

Ln221: How were integrons detected? This information is not available in M & M section.

Response: We have added the details of integrons detection in Ln137-138 in the revised manuscript.

Ln231: How were plasmids detected? This information is not available in M & M section.

Response: We have added the tool used to detect plasmid replicon in Ln139-140 in the revised mansucript.

Ln233-237: The replicon types such as IncII, IncFII, IncN, IncP6, col440II and IncQ were not found in table 2.

Response: Plasmid replicon information has been updated in Table 2 in the revised manuscript.

Ln240-241: Please provide the complete legend for Fig.1 Example what does arrows indicated?

Response: We have added the legends for Figure 1 in the revised manuscript.

Ln252: The total number of unique genes 2,324 is correct? From Fig.2 they are 2,517. Please make changes. 

Response: We have changed the total number of unique gene in Figure 2 in Ln256-257 in the revised manuscript.

Ln255: Please include how floral diagram is created in the M & M section?

Response: We have included the tool used to create floral diagram in the revised manuscript.

Ln265: Please remove’(ref)’

Response: We have added the reference in line 265 in the revised manuscript.

Ln266: I could see only 22 COG functions in figure 3.

Response: We have given details of 22 cog categories because among all the COG categories, chromatin structure and dynamics (B) category was not present in all the Salmonella serovars.

Ln269-270: Suggestions to change Y-axis as percentages and keep only most abundant functions (may be top 10 + combining all other functions).

Response: We have re structured the figure and and included the top most abundant functional group and combined all other functional group as suggested by the Reviewer in the revised manuscript.

Ln273: Please include number of genes near “highest and lowest”.

Response: We have included the number of genes in Ln277 as suggested by the Reviewer in the revised manuscript.

Ln275: Expand ‘E’ completely.

Response: We have expanded ‘E’ in Ln279 in the revised manuscript.

Ln288: Please include exact percentage values for G and L.

Response: The percentage value has been added in Ln 295-296 in the revised manuscript.

Ln293-328: These results are based on supplementary tables and these are presented for each isolate and it is difficult to compare the results. A comprehensive information like table 2 will help to look over the results clearly.

Response: All the virulence data table will be too lengthy to include, however, the genes which were not detected in all the serovars are included in Table 2.

Ln357-359: Fig.5 results were not discussed in results section.

Response: The combined results were discussed in the results section in Ln349-360.

Ln362: This is ten right?

Response: We have detected resistance and virulence genes in ten isolates, but only eight isolates harbored at least one virulence gene on phage. 

Discussion

Discussion is well organized however it is too lengthy, restatement of data (restatement of results) observed in some places.

Response: The authors have restructured the sentences to avoid redundancy and shorten the revised manuscript. 

References

Too many references.

Response: Some of the references have been deleted in the revised manuscript. However, we have still included many as they are important compare the findings in our study to previous work in the literature. 

Figures

Except figure 2, none of the figures are in good quality they are not clearly visible for interpretation, in addition more information is needed to add in the legend for easy interpretation. 

Response: We have modified the figures for clarity. We have submitted the high quality figures with 300 DPI as per manuscript requirements. 

Reviewer #2: 

The manuscript reports an extensive genome analysis of 14 diverse Salmonella serotypes isolated from swine using whole genome sequencing. The results suggest that majority of the genes were part of core genome and dispensable genes were acquired for adaptability. Here, the authors have performed analysis using various bioinformatics tools to identify and predict genetic repertoire in salmonella serovars. I found this paper of broad relevance to the field, especially to antibiotic resistance and public health. However, I have provided general comments and suggestions below that are necessary to improve the manuscript.

Minor comments

1. The authors have stated high concordance between phenotypically confirmed resistances and detected ARGs, however they did not state why it was not well correlated. The authors need to explain.

Response: We would like to thank the reviewer for this comment. The panel which we have used for phenotypic resistance assay contains only seventeen antimicrobials and does not include trimethoprim, chloramphenicol and fosfomycin. Hence, the authors have not included the genes conferring resistance to these antimicrobials for the correlation analysis. However, we have described rest of the resistance genes and their corresponding phenotypic resistance in the results and discussions section. We noticed high concordance between predicted resistance and detected phenotype. 

2. The authors noticed two distinct pattern of CRISPR-cas3 gene, how did they validated the findings?

Response: Our analysis is based on the previous studies reporting two different cas 3 genes in Salmonella. We have aligned all the cas3 genes and constructed the phylogenetic tree which clearly clustered these isolates into two different clads.

3. The authors have showed clusters of orthologous genes in core and accessory genes (figure 4). Authors might also consider to show the distribution of cogs in unique genes.

Response: We have included the clusters of orthologous genes in unique genes and modified the figure and text accordingly.

4. Authors might consider to add a figure showing the distribution of pan genome (core genes, accessary genes and unique genes) content in each salmonella serotypes (for eg: serovars in x-axis and number of genes in y-axis).

Response: We have added the additional supplementary figure (S1 Figure) showing the distribution of pan genome (core genes, accessary genes and unique genes) content in each Salmonella serotypes. 

5. The authors need to elaborate the materials and methods sections.

Response: The materials and methods section have been elaborated for more clarity in the revised manuscript.

6. Authors should add cogs database reference (line 265)

Response: The authors would like to thank the reviewer for pointing out this. The COG database reference has been included in Ln269 in the revised manuscript.

---

## [Decision Letter · Decision Letter 1]

16 Oct 2019

Genomic comparison of diverse Salmonella serovars isolated from swine

PONE-D-19-17181R1

Dear Dr. Frye,

We are pleased to inform you that your manuscript has been judged scientifically suitable for publication and will be formally accepted for publication once it complies with all outstanding technical requirements.

With kind regards,

Yung-Fu Chang

Academic Editor

PLOS ONE

Additional Editor Comments (optional):

Reviewers' comments:

Reviewer's Responses to Questions

**Comments to the Author**

1. If the authors have adequately addressed your comments raised in a previous round of review and you feel that this manuscript is now acceptable for publication, you may indicate that here to bypass the “Comments to the Author” section, enter your conflict of interest statement in the “Confidential to Editor” section, and submit your "Accept" recommendation.

Reviewer #1: All comments have been addressed

Reviewer #2: All comments have been addressed

2. Is the manuscript technically sound, and do the data support the conclusions?

Reviewer #1: Yes

Reviewer #2: Yes

3. Has the statistical analysis been performed appropriately and rigorously? 

Reviewer #1: N/A

Reviewer #2: N/A

4. Have the authors made all data underlying the findings in their manuscript fully available?

Reviewer #1: Yes

Reviewer #2: Yes

5. Is the manuscript presented in an intelligible fashion and written in standard English?

Reviewer #1: Yes

Reviewer #2: Yes

6. Review Comments to the Author

Reviewer #1: Thr authors have addressed all the comments and substantially improved the manuscript. I have not further comments.

Reviewer #2: All comments are addressed well in the manuscript.

An extensive genome analysis of 14 diverse Salmonella serotypes isolated from swine using whole genome sequencing was reported well. Further the authors have included a figure showing the distribution of pan genome (core genes, accessary genes and unique genes) for the readers to understand the genome repertoire of the salmonella serotypes

The authors also modified figure 4 to include clusters of orthologous genes in unique genes, as suggested. The manuscript is well written in standard English and the authors have presented a good scientific research article.

I recommend for accepting the manuscript.

7. PLOS authors have the option to publish the peer review history of their article (what does this mean?). If published, this will include your full peer review and any attached files.

Reviewer #1: No

Reviewer #2: No

---

## [Editor Report · Acceptance letter]

23 Oct 2019

PONE-D-19-17181R1 

Genomic comparison of diverse *Salmonella* serovars isolated from swine 

Dear Dr. Frye:

I am pleased to inform you that your manuscript has been deemed suitable for publication in PLOS ONE. Congratulations! Your manuscript is now with our production department. 

With kind regards,

on behalf of

Dr. Yung-Fu Chang 

Academic Editor

PLOS ONE